# High-quality histochemistry, immunohistochemistry, and immunofluorescence on xylene- and formalin-free paraffin-embedded tissues

Maarten Niemantsverdriet, Anne Mei Eshuis, Naomi van der Horst, Rory Ogink, Pien van Smeerdijk, Jose van der Starre-Gaal, Agnes Marije Hoogland [ID]*

Isala pathology, Dr. Van Heesweg 2, 8025 AB Zwolle, The Netherlands

* a.m.hoogland@isala.nl

## Abstract

Pathology is a vital part of modern medicine, playing an essential role in disease diagnosis, prognosis, and therapy choice. The field of pathology depends on traditional tissue embedding techniques, still routinely using toxic and even carcinogenic substances to process tissues for diagnostics. Chronic exposure of humans to xylene and/or formalin, two essential components in the process, may result in severe health problems. Accumulating insights resulted in a ban of formalin in Europe for multiple applications and tight restrictions for almost all other uses since its toxicity levels were upgraded to carcinogen grade 1B and mutagen grade 2. A remarkable exception to these regulations is the use of formalin in the field of pathology, because of the existing lack of formalin-free tissue embedding methods that produce high-quality pathology specimens. Here we show that a novel embedding technique employing supercritical Carbon dioxide ($CO_2$), used in a xylene- and formalin-free manner, results in high-quality tissue samples, suitable for the fundamental techniques used in pathology: histochemistry, immunohistochemistry, and immunofluorescence.

## Introduction

Pathology is fundamental for disease diagnosis, prognostic stratification, and therapeutic choice. The field of pathology was revolutionized by the application of microscopy to the study of diseased tissues from about 1800. Three current cornerstones of solid tissue pathology comprise histochemistry, immunohistochemistry, and immunofluorescence [1]. Histochemistry studies the chemistry of tissue sections by microscopy after they have been treated with specific reagents (dyes) to visualize the features of tissues and individual cells [2]. Immunohistochemistry (IHC) is a laboratory technique used to detect specific cellular or extracellular antigens based on antibody recognition, covering a huge number of cell and tissue types at different stages of differentiation. IHC detects targets via a label attached directly or indirectly

**Data availability statement:** All relevant data are within the paper and its Supporting Information files.

**Funding:** The author(s) received no specific funding for this work.

**Competing interests:** I have read the journal's policy and the authors of this manuscript have the following competing interests: Agnes Marije Hoogland is included as one of the inventors in the Tispa patent WO 2024/117901 A1 of Tispa Medical B.V. and Stichting Isala Klinieken on the Non-Fixed Paraffin-Embedded (NFPE) technique, with additional patents pending. The other authors do not report a conflict of interest. This does not alter our adherence to PLOS ONE policies on sharing data and materials.

to the antibody, which has to be visualized by a staining reaction. The type of label specifies the chemistry required to develop and observe the antigen-antibody reaction [3]. Many of these antibodies are highly specific to a cell type or organ, providing great diagnostic value in (tumour) tissue diagnostics [4]. Immunofluorescence (IF) may employ fluorochromes directly conjugated to the antibody (direct immunofluorescence or DIF), allowing direct visualization by using a fluorescence microscope [4]. This approach is specifically suitable for the analysis of fresh frozen tissues, but its use is specific for a subset of diseases, mainly in skin for autoimmune blistering diseases and other dermatoses [5], and in kidney for glomerular diseases [6]. Next to these three "traditional" cornerstones of pathology, in recent decades, the field of pathology has become increasingly dependent on molecular analysis for specific diagnosis, prognosis, and therapy choice [7].

Processing patient tissues is routinely done in pathology laboratories worldwide by formalin fixation and paraffin embedding (FFPE) with the intention of preserving tissue morphology and enabling immunohistochemical analysis [8,9]. This process uses xylene, which has been shown to cause toxic effects in immunological, gastrointestinal, respiratory, developmental, reproductive, and CNS systems [10], and formaldehyde which is very hazardous and can cause acute oral toxicity, acute dermal toxicity, acute inhalation toxicity, skin corrosion/irritation, serious eye damage/eye irritation, skin sensitization, germ cell mutagenicity, carcinogenicity, and specific target organ toxicity [11]. Formaldehyde has also been classified as a volatile organic compound (VOC), which are environmental pollutants that have been associated with stratospheric ozone depletion and global climate change [12]. Additionally, formalin fixation negatively impacts Deoxyribonucleic acid (DNA) quality and quantity and thereby potentially reduces the quality of diagnosis, prognosis, and therapy choice based on molecular analysis [13–15]. Based on accumulating insights, formalin was banned in Europe for multiple applications, and tight restrictions were applied for almost all other uses since its toxicity levels were upgraded to carcinogen grade 1B and mutagen grade 2 [16–19]. A remarkable exception to these regulations is the use of formalin in the field of pathology, because of the existing lack of formalin-free tissue embedding methods that produce high-quality pathology specimens with histochemistry and immunohistochemistry results similar to formalin-fixed tissues [19–21].

With the intention to eliminate the use of the toxic solvent xylene from the tissue embedding process, a bold novel tissue processing method was developed, using supercritical Carbon dioxide ($CO_2$) as an intermediate [22]. The discovery of this technique has led to a sustainable processing method for diagnostic tissues. We recently published a study where we show that this method can also be used to process fresh, unfixed tissues, completely omitting formaldehyde from the process in addition to xylene [23]. Processing tissues in this xylene and formalin-free manner, Non-Fixed Paraffin-Embedded (NFPE) tissues, results in samples with higher quality DNA, potentially improving molecular analysis [23]. In the study presented here, we analysed whether NFPE processing also results in high-quality tissue samples, suitable for the fundamental techniques used in pathology: histochemistry, immunohistochemistry, and immunofluorescence.

## Methods and materials

### Tissue library

Samples were collected between 1-1-2022 and 30-12-2024. After receiving fresh tissue from the operating room at the Department of Pathology, the tissue was inspected by a pathologist, and the parts needed to make the diagnosis were designated, according to WHO and local protocols, after which the parts that could be appointed as residual tissue were assigned. Parts of the residual tissues were taken, anonymized, and used for tissue library building in this study. Aorta, appendix, colon, kidney, liver, lung, ovarian tube, pancreas, placenta, salivary gland, skin, testis, thymus, thyroid, tonsil, and uterus tissues were used.

### Ethics

The research approach of this study was submitted to the local METC (Medical Ethics Committee, Isala Zwolle, the Netherlands). The Daily Board of the METC Isala Zwolle has reviewed the research proposal. As a result of this review, the Committee informs that the rules laid down in the Medical Research Involving Human Subjects Act (also known by the Dutch abbreviation WMO) do not apply to this research. This decision was stated on January 15, 2018, and filed under file number 180107 in the local METC archives. The need for informed consent documentation was waived and all experimental protocols were approved as part of the ethics committee approval. This is also described in Niemantsverdriet et al [23].

### Tissue processing

For tissues used for histochemistry and immunohistochemistry stainings, one part was processed as the gold standard (formalin-fixed paraffin-embedded, FFPE) as described in Niemantsverdriet et al [23]. For NFPE, another part was directly, freshly processed in the TISPA I tissue processor; fresh, unfixed tissue was loaded in the vessels of the Tispa I tissue processor (Tispa Medical) and processed using the "Large sample" protocol. First, the system was prepared by pumping paraffin in buffer vessels of the machine (4 min., 60°C, 0 bar), pumping ethanol in buffer vessels of the machine (2 min., 60°C, 0 bar), bringing the (tissue)vessels to a pressure of 150 bar (3 min, 54°C, 150 bar) and keeping at a pressure of 150 bar (20 min., 55°C, 150 bar). Next, 3 cycles of EtOH/$CO_2$ gradient steps were applied (25 min each): system was prepared to pump ethanol to the (tissue) vessel (3 min., 56°C, 150 bar), ethanol 99−100% was pumped in the vessel holding the tissue (3 min., 57°C, 150 bar), after which $CO_2$ was pumped in, mixing ethanol and $CO_2$ (3 min., 58°C, 142 bar), dewatering the tissue by holding at pressure (15 min., 58°C, 142 bar) and pressure was raised again to 150 bar (1 min. 59°C, 150 bar). After the 3 cycles of EtOH/$CO_2$ gradients, fresh $CO_2$ was pumped in to rinse the tissues from ethanol (30 min, temp gradient 60−64°C, 150 bar). Next, $CO_2$ was pumped in at lower pressure (4 min., 64°C, 135 bar). Paraffin was pumped in (5 min., 64°C, 135 bar) and held at pressure (5 min. 64°C, 135 bar). After the paraffin step, the process was finalized by gradually depressurizing the vessels from 135 to 0 bar (18 min., 64°C, 135−0 bar) and a vacuum step (30 min, 64°C, −1 bar). Another part was processed as Xylene-free paraffin-embedded (XFPE) sample, as described in Bleuel et al [22]. In this study, XFPE tissue was subjected to 24 hours of fixation in buffered formalin and processed in the TISPA I tissue processor (Tispa medical) in the same way NFPE tissue was processed. In case any tissue was left, this was freshly frozen and stored at −80°C. Selected NFPE tissues were collected, of which 2 mm cores were used to generate a Tissue Microarray (TMA) for bulk staining. For assessment of immunofluorescence stainings, tonsil, colon, and thyroid tissues were used because they are known to intrinsically express two or more markers that are used for immunofluorescence. These tissues were split; one part was processed as non-fixed paraffin-embedded (NFPE), and another part was freshly frozen and stored at −80°C. For all paraffin blocks and frozen tissues used for this study, 3μm sections were cut. Basic hematoxylin and eosin (HE) staining was performed on all samples for general evaluation of tissue quality and morphology.

## Histochemistry and immunohistochemistry

A set of internationally widely used histochemical and immunohistochemical markers (Tables 1 and 2, respectively) was selected to be validated on the NFPE tissues, using conventional FFPE tissues as reference. Occasionally, XFPE material was also stained. Based on the types of tissues collected in the different tissue blocks of the library, blocks with suitable tissue types showing specific staining patterns were selected to test each marker (Tables 1 and 2). Stainings were performed on 3μm sections. The starting point for the staining protocol used for each marker was the staining protocol currently used for diagnostics, optimized for FFPE tissues. When needed, steps of the protocol were adjusted to get optimal staining patterns for NFPE staining (Table 1 for histochemistry and Table 2 for immunohistochemistry). The staining methods used and adjustments were as briefly mentioned in Table 1 for histochemistry and Table 2 for immunohistochemistry. Detailed methods are omitted because of page space and because to reproduce these results in other labs, the stainings on NFPE tissues would have to be optimized, starting from the lab's own standard protocols used for diagnostics and not from our protocol. For histochemistry (Fig 1, 2, S1, Tables 1 and 3) and immunohistochemistry (Fig 3, S2 and S3, and Tables 2 and 4), the test performance characteristics were based on the level of basic analytical sensitivity and specificity, the technical quality of morphology, signal-to-noise ratio, and counterstaining [24]. For all final stainings, a technical replicate was performed on each tissue type, and biological replicates are indicated by the different tissues in Tables 3 and 4. The assessments of all stainings were initially performed blinded by an expert reviewer (specialized technician), followed by a not-blinded assessment by a general pathologist, and finally verified by a specialized pathologist. Inter-observer agreement of the final stainings was 100% for histochemistry (Table 3) and immunohistochemistry (Table 4). For quantification of P63 in NFPE, XFPE, and FFPE, and Ker5 in NFPE in supplementary figure S5 Fig, 10 sets of 4 adjacent cells with a positive staining pattern were selected throughout the image, and the staining intensity was quantified using

**Table 1. Used histochemistry stainings, methods, and adjustments for NFPE.**

| Histochemistry | | | |
|---|---|---|---|
| *Marker* | *Target* | *Method* | *Adaptations for NFPE to standard FFPE protocol* |
| HE (Hematoxylin-Eosin) | all tissues, basic stain | DAKO Cover Stainer HE protcol | Hematoxylin 30 seconds and 50% concentration of Eosin, 60 seconds |
| Alcian blue | cartilage, mucineus tissue | Benchmark Special Stains, Alcian Blue staining kit | incubation in blue 60 minutes |
| Azan | connective tissue | Benchmark Special Stains, Trichrome staining kit | incubation in red 16 minutes |
| Elastica von Giesson | connective tissue | Benchmark Special Stains, Elastica von Giesson staining kit | no adaptations |
| Giemsa | mestcells, nuclear details | Staining by hand | no adaptations |
| Hale's (colloidal) iron | mucinous tissue | Staining by hand | 15 min. incub. in potassium ferrocyanide-hydrochloric acid and incub. in red 5 min. |
| Masson Goldner (trichrome) | connective tissue, muscle cells | Benchmark Special Stains, Green for Trichrome Staining kit | no adaptations |
| Periodic Acid Schiff (PAS) | glycogen, nuclear details | DAKO Cover Stainer PAS protocol | no adaptations |
| Periodic Acid Schiff-Diastase (PAS-D) | glycogen, after diastases, nuclear details | DAKO Cover Stainer PAS-D protocol | no adaptations |
| Reticulin | reticulin fibres | Benchmark Special Stains, Reticulum II staining kit | Slide dipped in formalin for ten minutes following deparaffinization |
| Schmorl | melanin pigment | Staining by hand | no adaptations |

**Table 2. Used immunohistochemistry stainings, methods and adjustments for NFPE.**

| Immunohistochemistry | | | |
|---|---|---|---|
| Marker | Target | Method | Adaptations for NFPE to standard FFPE protocol |
| CD1a | thymus, Langerhanscells, T-ALL | Roche, RTU, clone: CD1a (EP3622) Rabbit Monoclonal | CC1 buffer 16 minutes, incubation time 12 minutes |
| CD2 | T-lymfocytes | Roche, RTU, clone: CD2 (MRQ-11) | no adaptations |
| CD3 | T-lymfocytes | Roche RTU, clone: CD3 (2GV6) | CC1 buffer 40 minutes |
| CD4 | T-helper cells, NK cells | Roche, RTU, clone: CD4 (SP35) | CC1 buffer 48 minutes, incubation time 28 minutes |
| CD5 | T-lymfocytes | Roche, RTU, clone: CD5 (SP19) | no adaptations |
| CD8 | Cytotoxic T-cells, NK cells | Roche, RTU, clone: CD8 (SP57) | no adaptations |
| CD20 | B-lymfocytes | Roche RTU, clone: CD20 (L26) | no adaptations |
| CD34 | blasts in bone marrow, endothelium | Roche, RTU, clone: CD34 (QBEnd/10) | no adaptations |
| CD68 | macrophages | Roche RTU, clone: CD68 (KP-1) | **not satisfactory** |
| CD79a | B-lymfocytes/ plasmacells | Roche, RTU, clone: CD79a (SP18) Rabbit Monoclonal | no adaptations |
| CD138 | plasmacells | Roche, RTU, clone: CD138 (B-A38) | no adaptations |
| CDX2 | epithelium of lower gastro-intestinal tract | Roche, RTU, clone: CDX-2 (EPR2764Y) | no adaptations |
| Cytokeratin 5/ Keratin 5 | epithelium, squamous epithelium, urothelium, mesothelium | Roche, RTU, clone: CK5 (SP27) | no adaptations |
| Cytokeratin 7/ Keratin 7 | epithelium of mamma, lung, upper gastric tract, mesothelium | Roche, RTU, clone: CK7 (SP52) Rabbit Monoclonal | CC1 buffer 24 minutes, incubation time 16 minutes |
| Cytokeratin 8/18/ Keratin 8/18 | non-squamous epithelium/ adenocarcinoma of stomach/colon/oesophagus | Roche, RTU, clone: CK8 & 18 (B22.1 & B23.1) | CC1 buffer 8 minutes |
| Cytokeratin 20/ Keratin 20 | epituelium of lower intestinal tract, urothelium | Roche, RTU, clone: CK20 (SP33) Rabbit Monoclonal | incubation time 8 minutes |
| Cytokeratin AE1/AE3 (pankeratin) | epithelium | Roche, RTU, clone: cytokeratin (AE1/AE3) | no adaptations |
| e-cadherin | epithelium | Roche, RTU, clone: E Cadherin 36 | no adaptations |
| Ki67 | proliferating cells | Roche, RTU, clone: KI67 (30−9) | no adaptations |
| MLH1 | mismatch repair protein, nucleus | Roche, RTU Leica BOND, clone: MLH1 (ES05) | CC1 88 minutes, incubation time 60 minutes, amplifier 4 minutes |
| MSH2 | mismatch repair protein, nucleus | Roche, RTU, clone: MSH2 (G219-1129) | CC1 96 minutes, incubation time 76 minutes |
| MSH6 | mismatch repair protein, nucleus | Roche, RTU, clone: MSH6 (SP63) | CC1 92 minutes, incubation time 52 minutes |
| OCT4/ POU5F1 | germinal cells, seminoma | Roche, RTU, clone: OCT4 (MRQ-10) | CC1 32 minutes, incubation time 24 minutes |
| P53 | tumorsuppressor protein | Roche, RTU, clone: P53 (DO-7) | no adaptations |
| P63 | epithelium | Roche, RTU, clone: P63 (4A4) | incubation prolonged with 24 minutes and amplifier added to standard protocol |
| PAX-8 | epithelium of gynocologic origin and kidney | Roche RTU, clone: PAX-8 (SP348) | **not satisfactory** |
| PLAP | placental tissue, germinal cells, seminoma | Roche, RTU, clone: PLAP (NB10) | no adaptations |
| PMS-2 | mismatch repair protein, nucleus | Roche, RTU, clone: PMS2 (EP51) | Ab incubation 10 min shorter |
| SOX-10 | sweat gland, nerve tissue and melanocytic cells | Roche RTU, clone: SOX-10 (SP267) | no adaptations |

*(Continued)*

**Table 2.** (Continued)

| Immunohistochemistry | | | |
|---|---|---|---|
| TDT | precursor B- and T- Lymfocytes, ALL | Roche, RTU, clone: TDT Rabbit Polyclonal | **not satisfactory** |
| TTF1 | epithelium of lung and thyroid gland | Roche RTU, clone: TTF-1 (SP141) | incubation time 16 minutes |

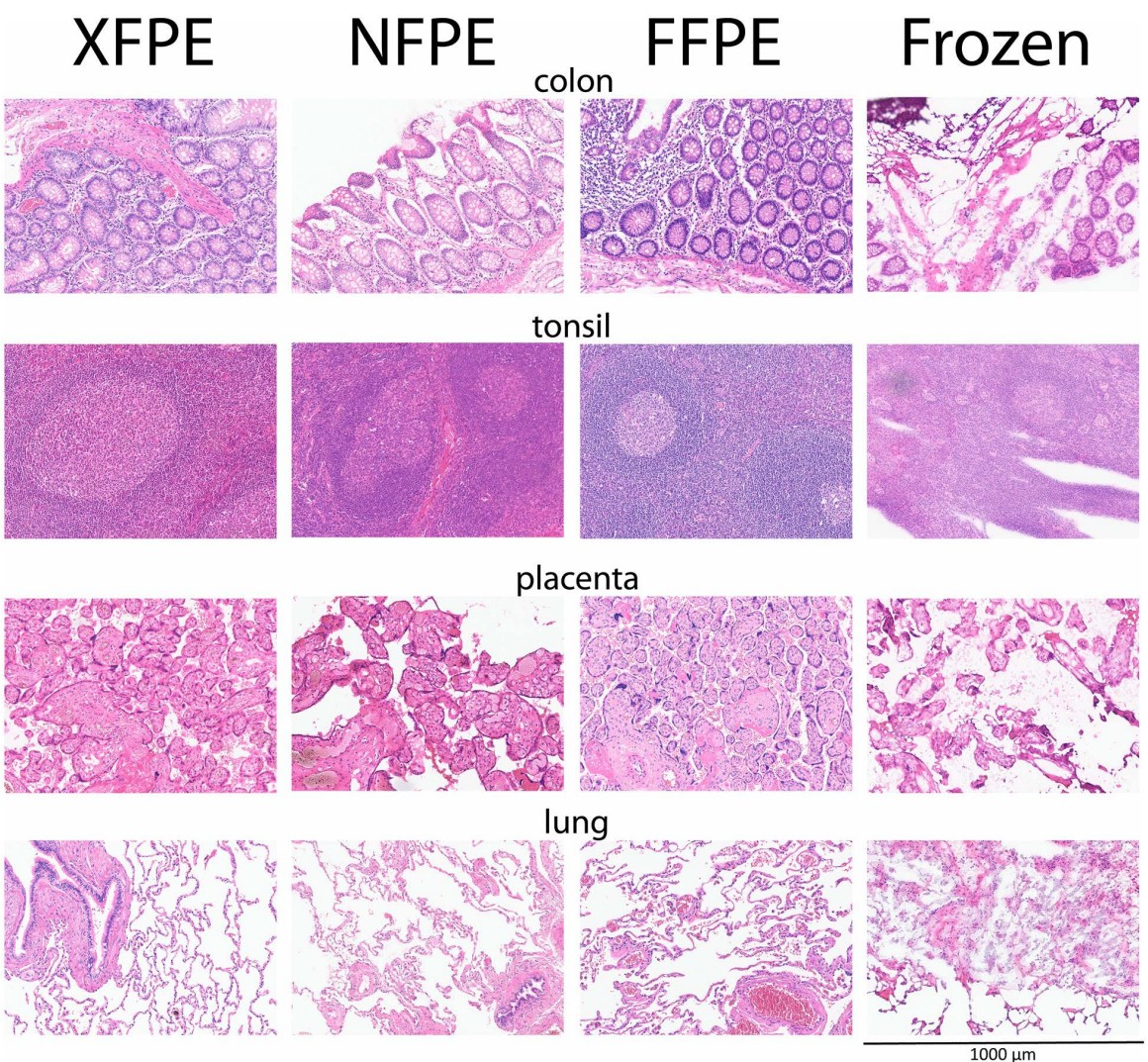

**Fig 1. Representative Hematoxylin and Eosin (HE) stainings on Xylene-free Paraffin-Embedded (XFPE), Non-fixed Paraffin-Embedded (NFPE), Formalin-Fixed Paraffin-Embedded (FFPE), and Frozen tissues.** All pictures have the same width (1000 μm).

ImageJ software. Average and standard deviation were calculated, and a graphical image was created in Excel. Significance levels were determined using a student's T-test.

For three immunohistochemistry markers, CD68, PAX-8, and TDT, all attempted adaptations to the standard protocol did not result in specific staining with acceptable staining intensity and acceptable background levels (Table 4 and S2 and S3).

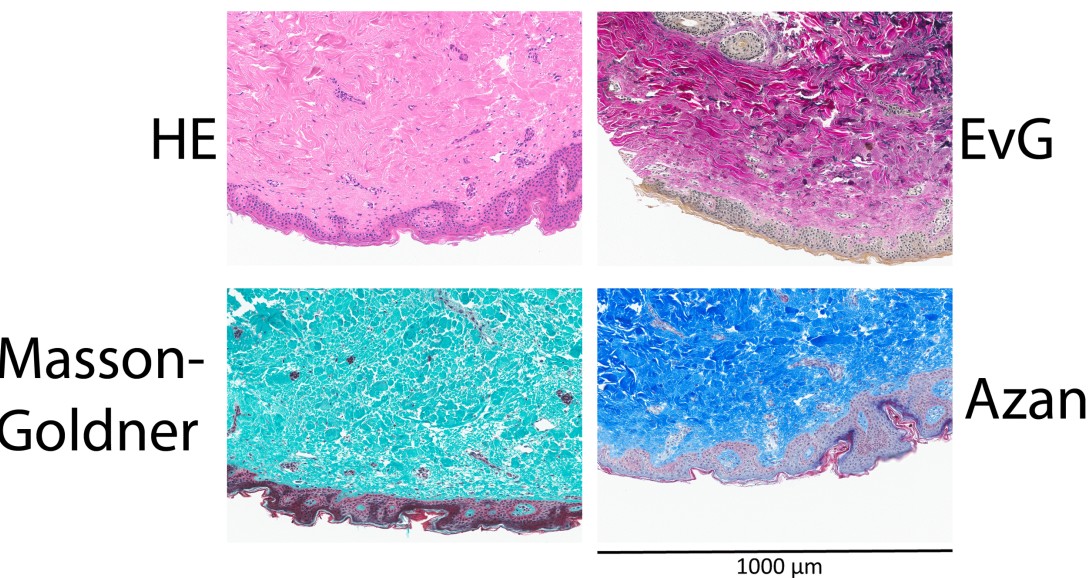

**Fig 2. Skin morphology stainings on Non-fixed Paraffin-Embedded (NFPE) tissues.** HE; Hematoxylin and Eosin (HE) staining, EvG; Elastica von Gieson, Masson-Goldner, and Azan staining. All pictures have the same width (1000 μm).

### Immunofluorescence

An internationally acknowledged set of immunofluorescence markers for skin [5] and kidney [6] tissues were validated on NFPE tissues and compared to fresh frozen tissues (Table 5, Fig 5, and S4). For the evaluation of immunofluorescence stainings (Fig 5, S4, and Table 5), control tissues, tonsil, thyroid, and colon, were used to optimize the staining protocols for NFPE tissues. For the different markers, the immunofluorescence properties of the control tissues were used to adjust the protocols as discussed in the manufacturer's manuals. The staining methods used and adjustments were as briefly mentioned in Table 6. Detailed methods are omitted because to reproduce these results in other labs, the stainings on NFPE tissues would have to be optimized, starting from the labs' own standard protocols and not from our protocol. For all final stainings, a technical replicate was performed on each tissue, and biological replicates are as indicated by different tissues in Table 5. The assessments of all stainings were initially performed blinded by an expert reviewer (specialized technician), followed by a not-blinded assessment by a general pathologist, and finally verified by a specialized pathologist. Inter-observer agreement of the final stainings was 100% for immunofluorescence stainings (Table 5).

### Results

Tissue was collected from aorta, appendix, colon, kidney, liver, lung, ovarian tube, pancreas, placenta, salivary gland, skin, testis, thymus, thyroid, tonsil, and uterus. Formalin-fixed-paraffin-embedded (FFPE), Xylene-free Paraffin-embedded (XFPE), Non-Fixed Paraffin-Embedded (NFPE), and/or Fresh Frozen tissues were generated. First, Hematoxylin and Eosin (HE) stainings were performed for general evaluation of tissue quality and morphology. HE staining was optimized for NFPE tissues. No tissue processing-induced differences were observed in HE staining between the FFPE tissues, XFPE, and NFPE tissues (Fig 1 and S1). In contrast, a major improvement was observed in morphology in the HE staining of NFPE slides compared to the frozen section material designated to evaluate immunofluorescence markers (Fig 1 and S1). Eleven histochemistry stainings (Figs 1, 2, 3, S1, and Table 3) and thirty-one immunohistochemistry stainings (Fig 4, S2 and S3, and Table 4) were performed on NFPE, FFPE, and occasionally XFPE

**Table 3. Histochemistry stainings, tissues used, and results.**

| Histochemistry | Tissue | XFPE | NFPE | FFPE | Frozen | results NFPE | ER | GP | SP | Figure |
|---|---|---|---|---|---|---|---|---|---|---|
| HE (Hematoxylin-Eosin) (11) | colon | | | | | specific, no background, good morphology | | | | Fig 1 |
| | placenta | | | | | | | | | Fig 1 |
| | lung | | | | | | | | | Fig 1 |
| | tonsil | | | | | | | | | Fig 1 |
| | skin | N.A. | | N.A. | | | | | | Figs 2 and S1 |
| | uterus | | | N.A. | N.A. | | | | | S1 Fig |
| | aorta | N.A. | | N.A. | N.A. | | | | | S1 Fig |
| | mamma | | | N.A. | N.A. | | | | | S1 Fig |
| | salivary gland (nil) | N.A. | | N.A. | N.A. | | | | | S1 Fig |
| | kidney | N.A. | | N.A. | N.A. | | | | | S1 Fig |
| | thyroid | N.A. | | N.A. | | | | | | S1 Fig |
| Alcian blue (4) | salivary gland | N.A. | | N.A. | N.A. | specific, no background, good morphology | | | | S1 Fig |
| | appendix | N.A. | | N.A. | N.A. | | | | | S1 Fig |
| | aorta | N.A. | | N.A. | N.A. | | | | | S1 Fig |
| | colon | N.A. | | N.A. | N.A. | | | | | S1 Fig |
| Azan (3) | colon | | | | N.A. | specific, no background, good morphology | | | | Fig 3 |
| | aorta | N.A. | | N.A. | N.A. | | | | | S1 Fig |
| | skin | N.A. | | N.A. | N.A. | | | | | Fig 2 |
| Elastica von Gieson (4) | uterus | N.A. | | N.A. | N.A. | specific, no background, good morphology | | | | S1 Fig |
| | lung | N.A. | | N.A. | N.A. | | | | | S1 Fig |
| | colon | N.A. | | | N.A. | | | | | S1 Fig |
| | skin | N.A. | | N.A. | N.A. | | | | | Fig 2 |
| Giemsa (2) | colon | N.A. | | N.A. | N.A. | specific, no background, good morphology | | | | S1 Fig |
| | skin | N.A. | | N.A. | N.A. | | | | | S1 Fig |
| Hale's (colloidal) iron (4) | lung | N.A. | | N.A. | N.A. | specific, no background, good morphology | | | | S1 Fig |
| | aorta | N.A. | | N.A. | N.A. | | | | | S1 Fig |
| | kidney | N.A. | | N.A. | N.A. | | | | | S1 Fig |
| | placenta | N.A. | | N.A. | N.A. | | | | | S1 Fig |
| Masson Goldner (tri-chrome) (3) | aorta | | | | N.A. | specific, no background, good morphology | | | | Fig 3 |
| | ovarian tube | N.A. | | N.A. | N.A. | | | | | S1 Fig |
| | skin | N.A. | | N.A. | N.A. | | | | | Fig 2 |
| Periodic Acid Schiff (PAS) (3) | pancreas | N.A. | | N.A. | N.A. | specific, no background, good morphology | | | | S1 Fig |
| | kidney | N.A. | | N.A. | N.A. | | | | | S1 Fig |
| | salivary gland (nil) | N.A. | | N.A. | N.A. | | | | | S1 Fig |
| Periodic Acid Schiff-Diastase (PAS-D) (3) | salivary gland | | | | N.A. | specific, no background, good morphology | | | | Fig 3 |
| | pancreas | N.A. | | N.A. | N.A. | | | | | S1 Fig |
| | kidney | N.A. | | N.A. | N.A. | | | | | S1 Fig |

*(Continued)*

**Table 3.** (Continued)

| Histochemistry | Tissue | XFPE | NFPE | FFPE | Frozen | results NFPE | ER | GP | SP | Figure |
|---|---|---|---|---|---|---|---|---|---|---|
| Reticulin (3) | placenta | N.A. | | N.A. | N.A. | specific, no background, good morphology | | | | S1 Fig |
| | kidney | N.A. | | N.A. | N.A. | | | | | S1 Fig |
| | skin | N.A. | | N.A. | N.A. | | | | | S1 Fig |
| Schmorl (1) | skin | N.A. | | N.A. | N.A. | specific, no background, good morphology | | | | S1 Fig |
| | | | | | | inter-observer agreement | 123/123*100= | | | 100% |

ER = Expert Reviewer, GP = General Pathologist, SP = Specialized Pathologist. Observer agreement with the conclusion shown in "results NFPE" is indicated in gray. Biological replicates are given in brackets behind the name of the staining. For all stainings two technical replicates were performed on each tissue type.

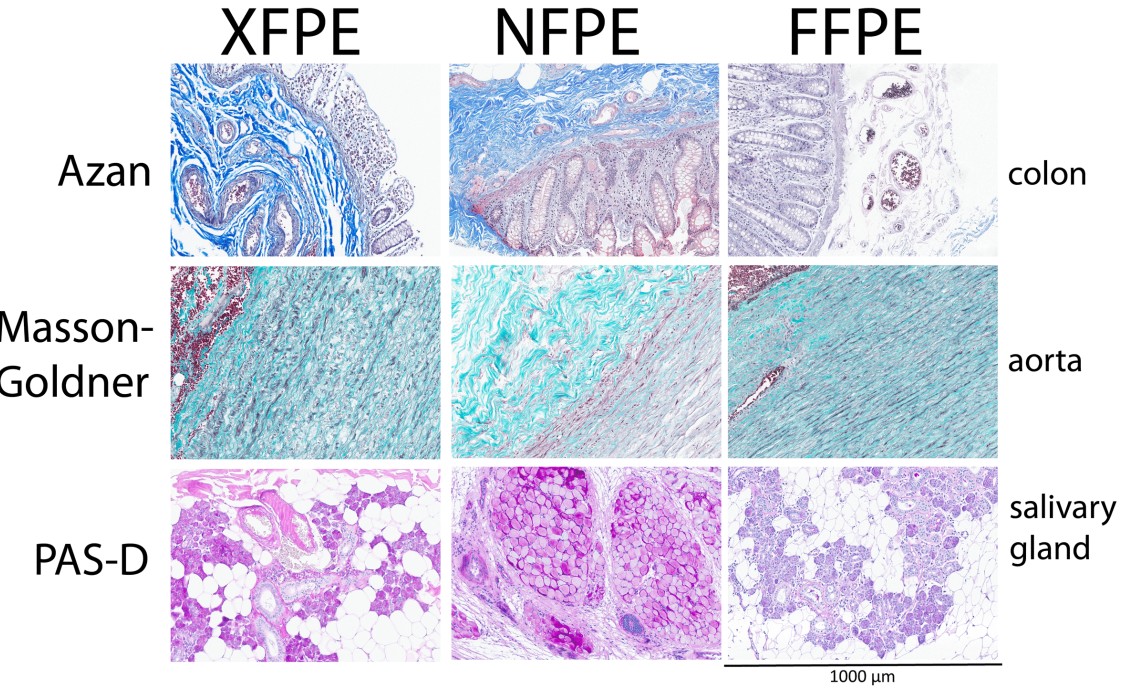

**Fig 3. Representative histochemical stainings on Xylene-free Paraffin Embedded (XFPE), Non-fixed Paraffin Embedded (NFPE), and Formalin-Fixed Paraffin Embedded (FFPE) tissues.** All pictures have the same width (1000 μm).

processed tissues. Seven direct immunofluorescence stainings (Table 5, Fig 5, and S4) were performed on Fresh Frozen and NFPE tissues.

The preserved morphology in NFPE samples observed in HE stainings (Figs 1 and S1) was emphasized after staining NFPE skin tissues for differences in collagen structure between the papillary dermis and the reticular dermis. In the papillary dermis the loose connective tissue contains less elastic fibres compared to the reticular dermis where collagen is more dense and elastin fibres are thick and conspicuous, this was visualized in Fig 2 where Hematoxylin and Eosin (HE) staining, Azan, Elastica van Gieson (EvG) and Masson-Goldner stainings show that these different qualities and quantities of fibers are preserved in NFPE skin tissues.

**Table 4. Immunohistochemistry stainings, tissues used, and results.**

| Immunohistochemistry a | Tissue | XFPE | NFPE | FFPE | results NFPE | ER | GP | SP | Figure |
|---|---|---|---|---|---|---|---|---|---|
| CD1a (2) | thymus | N.A. | | N.A. | specific, no background, good morphology | | | | S2 and S3 Figs |
| | skin | N.A. | | N.A. | | | | | S2 and S3 Figs |
| CD2 (3) | tonsil | N.A. | | | specific, no background, good morphology | | | | S2 and S3 Figs |
| | appendix | N.A. | | | | | | | S2 and S3 Figs |
| | duodenum | N.A. | | N.A. | | | | | S2 and S3 Figs |
| CD3 (3) | tonsil | N.A. | | N.A. | specific, no background, good morphology | | | | S2 and S3 Figs |
| | appendix | N.A. | | N.A. | | | | | S2 and S3 Figs |
| | colon | N.A. | | N.A. | | | | | S2 and S3 Figs |
| CD4 (2) | tonsil | N.A. | | N.A. | Specific, little background, good morphology | | | | S2 and S3 Figs |
| | appendix | N.A. | | | | | | | S2 and S3 Figs |
| CD5 (2) | tonsil | N.A. | | | Specific, little background, good morphology | | | | S2 and S3 Figs |
| | appendix | N.A. | | | | | | | S2 and S3 Figs |
| CD8 (2) | tonsil | N.A. | | | specific, no background, good morphology | | | | S2 and S3 Figs |
| | appendix | N.A. | | | | | | | S2 and S3 Figs |
| CD20 (4) | tonsil | N.A. | | N.A. | specific, no background, good morphology | | | | S2 and S3 Figs |
| | appendix | N.A. | | N.A. | | | | | S2 and S3 Figs |
| | colon | N.A. | | N.A. | | | | | S2 and S3 Figs |
| | thymus | N.A. | | N.A. | | | | | S2 and S3 Figs |
| CD34 (3) | liver | N.A. | | N.A. | specific, no background, good morphology | | | | S2 and S3 Figs |
| | appendix | N.A. | | N.A. | | | | | S2 and S3 Figs |
| | colon | N.A. | | N.A. | | | | | S2 and S3 Figs |
| CD68 (2) | appendix | N.A. | | N.A. | **Not satisfactory** | | | | S2 and S3 Figs |
| | placenta | N.A. | | N.A. | | | | | S2 and S3 Figs |
| CD79a (2) | tonsil | N.A. | | N.A. | specific, no background, good morphology | | | | S2 and S3 Figs |
| | appendix | N.A. | | N.A. | | | | | S2 and S3 Figs |
| CD138 (2) | tonsil | N.A. | | | specific, no background, good morphology | | | | S2 and S3 Figs |
| | appendix | N.A. | | | | | | | S2 and S3 Figs |
| CDX2 (3) | appendix | N.A. | | N.A. | specific, no background, good morphology | | | | S2 and S3 Figs |
| | pancreas | N.A. | | N.A. | | | | | S2 and S3 Figs |
| | colon | N.A. | | N.A. | | | | | S2 and S3 Figs |
| Cytokeratin 5/ Keratin 5 (4) | tonsil | N.A. | | | specific, no background, good morphology | | | | S2 and S3 Figs |
| | skin | N.A. | | N.A. | | | | | S2 and S3 Figs |
| | ovarian tube | N.A. | | N.A. | | | | | S2 and S3 Figs |
| | liver | N.A. | | | | | | | S2 and S3 Figs |
| Cytokeratin 7/ Keratin 7 (4) | ovarian tube | N.A. | | N.A. | specific, no background, good morphology | | | | S2 and S3 Figs |
| | placenta | N.A. | | N.A. | | | | | S2 and S3 Figs |

*(Continued)*

| Immunohistochemistry a | Tissue | XFPE | NFPE | FFPE | results NFPE | ER | GP | SP | Figure |
|---|---|---|---|---|---|---|---|---|---|
| | tonsil | N.A. | | | | | | | S2 and S3 Figs |
| | liver | N.A. | | N.A. | | | | | S2 and S3 Figs |
| Cytokeratin 8/18/ Keratin 8/18 (5) | appendix | N.A. | | | moderately specific, no background, good morphology | | | | S2 and S3 Figs |
| | liver | N.A. | | N.A. | | | | | S2 and S3 Figs |
| | placenta | N.A. | | N.A. | | | | | S2 and S3 Figs |
| | colon | N.A. | | N.A. | | | | | S2 and S3 Figs |
| | tonsil | N.A. | | N.A. | | | | | S2 and S3 Figs |
| Cytokeratin 20/ Keratin 20 (2) | appendix | | | | specific, no background, good morphology | | | | Fig 4 |
| | colon | N.A. | | N.A. | | | | | S2 and S3 Figs |
| Cytokeratin AE1/AE3 (pan-keratin) (2) | tonsil | N.A. | | N.A. | specific, no background, good morphology | | | | S2 and S3 Figs |
| | colon | N.A. | | N.A. | | | | | S2 and S3 Figs |
| e-cadherin (2) | appendix | N.A. | | | specific, no background, good morphology | | | | S2 and S3 Figs |
| | tonsil | N.A. | | | | | | | S2 and S3 Figs |
| Ki67 (4) | tonsil | N.A. | | | specific, no background, good morphology | | | | S2 and S3 Figs |
| | appendix | N.A. | | | | | | | S2 and S3 Figs |
| | placenta | N.A. | | N.A. | | | | | S2 and S3 Figs |
| | skin | N.A. | | N.A. | | | | | S2 and S3 Figs |
| MLH1 (2) | tonsil | N.A. | | N.A. | Specific, little background, good morphology | | | | S2 and S3 Figs |
| | appendix | N.A. | | N.A. | | | | | S2 and S3 Figs |
| MSH2 (2) | tonsil | N.A. | | N.A. | Specific, little background, good morphology | | | | S2 and S3 Figs |
| | appendix | N.A. | | N.A. | | | | | S2 and S3 Figs |
| MSH6 (2) | tonsil | N.A. | | N.A. | Specific, little background, good morphology | | | | S2 and S3 Figs |
| | appendix | N.A. | | N.A. | | | | | S2 and S3 Figs |
| OCT4/ POU5F1 (1) | testis | N.A. | | | specific, no background, good morphology | | | | S2 and S3 Figs |
| P53 (4) | tonsil | N.A. | | N.A. | specific, no background, good morphology | | | | S2 and S3 Figs |
| | appendix | N.A. | | N.A. | | | | | S2 and S3 Figs |
| | ovarian tube | N.A. | | N.A. | | | | | S2 and S3 Figs |
| | placenta | N.A. | | N.A. | | | | | S2 and S3 Figs |
| P63 (2) | skin | | | | specific, no background, good morphology | | | | Fig 4 |
| | tonsil | N.A. | | | | | | | S2 and S3 Figs |
| PAX-8 (2) | uterus | N.A. | | N.A. | **not satisfactory** | | | | S2 and S3 Figs |
| | ovarian tube | N.A. | | | | | | | S2 and S3 Figs |
| PLAP (2) | placenta | N.A. | | N.A. | specific, no background, good morphology | | | | S2 and S3 Figs |
| | testis | N.A. | | N.A. | | | | | S2 and S3 Figs |
| PMS-2 (1) | tonsil | N.A. | | N.A. | specific, no background, good morphology | | | | S2 and S3 Figs |

*(Continued)*

**Table 4.** (Continued)

| Immunohistochemistry a | Tissue | XFPE | NFPE | FFPE | results NFPE | ER | GP | SP | Figure |
|---|---|---|---|---|---|---|---|---|---|
| SOX-10 (3) | skin | N.A. | | | specific, no background, good morphology | | | | S2 and S3 Figs |
| | uterus (nerves) | N.A. | | N.A. | | | | | S2 and S3 Figs |
| | mamma | N.A. | | N.A. | | | | | S2 and S3 Figs |
| TDT (2) | thymus | N.A. | | N.A. | **not satisfactory** | | | | S2 and S3 Figs |
| | tonsil | N.A. | | N.A. | | | | | S2 and S3 Figs |
| TTF1 (2) | thyroid | N.A. | | N.A. | specific, no background, good morphology | | | | S2 and S3 Figs |
| | lung | | | | | | | | Fig 4 |
| | | | | | inter-observer agreement | 234/234*100= | | | 100% |

ER = Expert Reviewer, GP = General Pathologist, SP = Specialized Pathologist. Observer agreement with the conclusion shown in "results NFPE" is indicated in gray. Biological replicates are given in brackets behind the name of the staining. For all stainings two technical replicates were performed on each tissue type.

## Histochemistry

Of the eleven histochemical markers (including the HE), six markers did not require any adaptations to the standard (FFPE) protocol to achieve specific staining with no or acceptable levels of background and good conservation of morphology in NFPE samples (Table 3, Figs 2, 3 and S1). For four markers, small changes to the protocol were needed to provide specific, background-free staining patterns with conservation of morphology (Table 3, Fig 2, 3, and S1). For one marker, reticulin, the standard protocol resulted in no staining at all. All formalin-free adaptations attempted resulted in no specific staining pattern. Dipping the slide in formalin for ten minutes after deparaffinization, before starting the staining protocol, resulted in a specific, background-free staining pattern with conservation of morphology (Tables 1 and 2 and S1). In the end, all histochemical stainings were acceptable on NFPE tissues.

## Immunohistochemistry

Of thirty-one immunohistochemical markers used, fifteen worked well without any adaptations to the standard (FFPE) protocol to achieve good results on NFPE (Table 4, Fig 4, and S2 and S3). For thirteen markers, a change in the CC1 buffer incubation time, a change in the antibody incubation time, or a combination of both resulted in highly specific, reliable staining patterns with no or an acceptable amount of background staining (Table 4, Fig 4, and S2 and S3).

Although immunohistochemical analysis is qualitative and subjective, which makes it less applicable for exact quantification [25], we managed to produce P63 stainings on NFPE, XFPE, and FFPE that show no statistical difference in staining intensity (S5).

Thus, for 28 of the 31 IHC Markers used, a good or acceptable result was generated on NFPE tissues. For the other three markers (CD68, PAX-8, and TDT), all attempted adaptations to the standard protocol did not result in specific staining with acceptable staining intensity and acceptable background levels in NFPE (Table 4 and S2 and S3).

## Immunofluorescence

For the immunofluorescence markers, stainings on frozen tissue were used as the gold standard and compared to NFPE samples (Table 5, Fig 5, and S4). Striking differences in how well-preserved cellular and structural details were in the NFPE tissues compared to frozen tissues were observed after comparing morphology in HE stainings (Fig 1 and S1). Known artifacts in structures of nuclear membranes, cellular membranes, and elastic membranes in blood vessels

**Table 5. Immunofluorescence stainings, tissues used, and results.**

| Immunofluorescense | Tissue | NFPE | Frozen | overal result NFPE | ER | GP | SP | Figure |
|---|---|---|---|---|---|---|---|---|
| IgA (3) | tonsil | | | Specific, little background, good morphology | | | | Fig 5 |
| | colon | | | | | | | S4 Fig |
| | thyroid | | | | | | | S4 Fig |
| IgG (3) | tonsil | | | Specific, little background, good morphology | | | | S4 Fig |
| | colon | | | | | | | S4 Fig |
| | thyroid | | | | | | | Fig 5 |
| IgM (3) | tonsil | | | Specific, little background, good morphology | | | | S4 Fig |
| | colon | | | | | | | S4 Fig |
| | thyroid | | | | | | | S4 Fig |
| C3c (3) | tonsil | | | Specific, little background, good morphology | | | | S4 Fig |
| | colon | | | | | | | S4 Fig |
| | thyroid | | | | | | | S4 Fig |
| C1q (3) | tonsil | | | Specific, little background, good morphology | | | | S4 Fig |
| | colon | | | | | | | S4 Fig |
| | thyroid | | | | | | | S4 Fig |
| kappa (3) | tonsil | | | Specific, little background, good morphology | | | | S4 Fig |
| | colon | | | | | | | S4 Fig |
| | thyroid | | | | | | | S4 Fig |
| lambda (3) | tonsil | | | Specific, little background, good morphology | | | | S4 Fig |
| | colon | | | | | | | Fig 5 |
| | thyroid | | | | | | | S4 Fig |
| | | | | inter-observer agreement | 63/63*100= | | | 100% |

ER = Expert Reviewer, GP = General Pathologist, SP = Specialized Pathologist. Observer agreement with the conclusion shown in "results NFPE" is indicated in gray. Biological replicates are given in brackets behind the name of the staining. For all stainings two technical replicates were performed on each tissue type.

appeared in frozen tissues; these appeared damaged and blurry (Fig 1 and S1). We did not observe any of these artifacts in the matching NFPE tissues (Fig 1 and S1). For three of the seven immunofluorescence markers tested (IgM, C3c, and C1q), no adaptations were needed to the standard protocols. For these markers, there was less general background staining in the different tissues in NFPE compared to the frozen sections. Staining patterns for these markers in the reference tissues were highly specific (Table 5, Fig 5, and S4). For the other four immunofluorescence markers, small adjustments to the standard protocols (Table 6) resulted in acceptable staining intensity, acceptable background levels, and good morphology (Table 5, Fig 5, and S4). In the end, all immunofluorescence stainings were acceptable on NFPE tissues.

## Discussion

The cornerstones of pathology: histochemistry, immunohistochemistry, immunofluorescence, and, in recent years, molecular pathology, are essential for disease diagnosis, prognostic stratification, and therapeutic choice of a large variety of

**Table 6. used immunofluorescence stainings, methods and adjustments for NFPE.**

| Immunofluorescense | | | |
|---|---|---|---|
| Marker | Target | Method | Adaptations for NFPE to standard FFPE protocol |
| IgA | lymfocytes in germinal center | Roche, RTU, FITC Anti-IgA Primary Antibody | incubation time 8 minutes |
| IgG | lymfocytes in germinal center | Roche, RTU, FITC Anti-IgG Primary Antibody | incubation time 24 minutes and after protocol, 15 minutes washing in NaCl |
| IgM | lymfocytes in germinal center | Roche, RTU, FITC Anti IgM Primary Antibody | no adaptations |
| C3c | membrana elastica in vessel walls | Roche, RTU, FITC Anti-C3 Primary Antibody | no adaptations |
| C1q | membrana elastica in vessel walls | Roche, RTU, FITC Anti-C1q Primary Antibody | no adaptations |
| kappa | lymfocytes and plasmacells | Roche, RTU, FITC Anti-Kappa Primary Antibody | 15 minutes washing in NaCl, before the staining step |
| lambda | lymfocytes and plasmacells | Roche, RTU, FITC Anti-Lambda Primary Antibody | 15 minutes washing in NaCl, before the staining step |

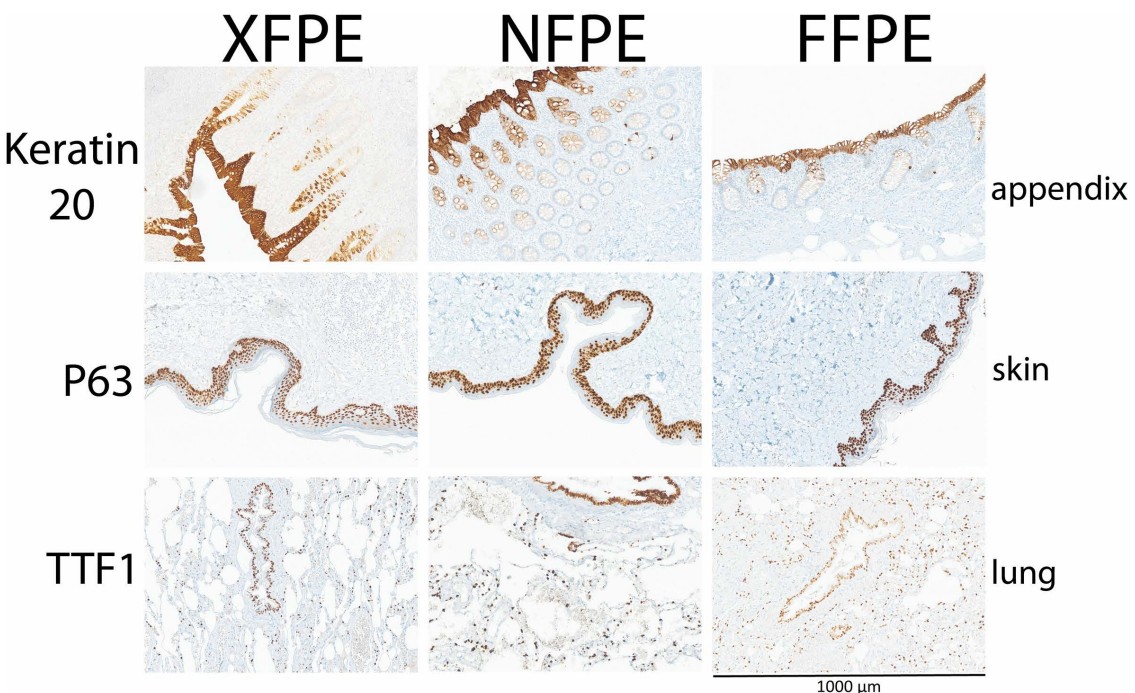

**Fig 4. Representative immunohistochemical stainings on Xylene-free Paraffin-Embedded (XFPE), Non-fixed Paraffin-Embedded (NFPE), and Formalin-Fixed Paraffin-Embedded (FFPE) tissues.** All pictures have the same width (1000 µm).

cancers and other diseases [1–5]. To perform these techniques, tissues need to be processed in a way suitable for staining and/or molecular analysis [1]– [5].

Formalin-fixation and paraffin-embedding of tissues is a standard procedure routinely performed in pathology laboratories all over the world, but it requires toxic xylene [10] and carcinogenic formaldehyde [11,13,16–19]. This technique is not up to today's safety standards, and formaldehyde was banned or tightly restricted in Europe for most applications [16–20]. Because of the existing lack of formalin-free tissue embedding methods that produce high-quality pathology specimens with histochemistry, immunohistochemistry, and immunofluorescence results similar to formalin-fixed tissues, the use of formaldehyde for tissue processing is still allowed in pathology labs in Europe [19–21].

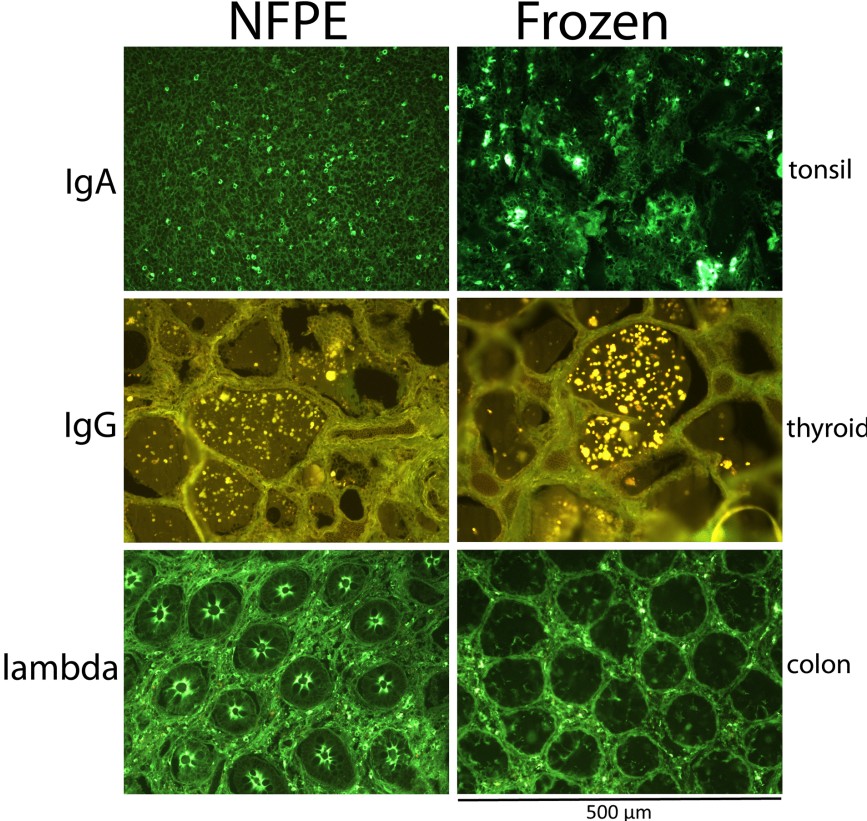

**Fig 5. Representative immunofluorescence stainings on Non-fixed Paraffin Embedded (NFPE) and Fresh Frozen tissues.** All pictures have the same width (500 μm).

We used a novel technique, Non-Fixed Paraffin Embedded (NFPE), to process tissues without formalin fixation and compared three cornerstones of solid tissue pathology: histochemistry, immunohistochemistry, and immunofluorescence stainings to the gold standard tissue processing techniques for these stainings. For histochemistry of the same tissues processed using the conventional gold standard of formalin-fixed and paraffin-embedded (FFPE) and Xylene-free paraffin-embedded (XFPE) tissues. Although XFPE has the benefit that toxic Xylene is not used and the processing time is shorter than for FFPE [23], the main focus of this study is on the comparison between FFPE and NFPE because FFPE is the current gold standard and, in addition to the benefits of XFPE, the main additional benefit of NFPE is that it is used without formaldehyde, which is carcinogenic and known to damage DNA [11,13,16–19]. In addition, we compared immunofluorescence stainings in NFPE tissues to its gold standard, frozen tissues. Because tissues lack homogeneity, already in unprocessed form, the resulting stainings also lack homogeneity [25], this is independent of tissue processing method. Therefore, for a general, novel study, testing a large quantity of histochemical, immuno-histochemical, and immunofluorescence stainings, thorough observational analysis is more suitable than statistical comparison. Our results were analyzed by three observers, and for all conclusions about staining results, inter-observer agreement was 100% (Tables 3–5).

All eleven histochemical and all seven immunofluorescence markers tested gave good (often better) results on NFPE tissues using the standard protocol or after minor modifications of standard protocols.

One of the histochemistry stainings, reticulin, did require the slide to be dipped in formalin (4% formaldehyde) for ten minutes after deparaffinization to give a good result. This is unfortunate considering our quest to completely eliminate

formaldehyde from the tissue analysis process. On the other hand, reticulin staining is needed only for liver and kidney diagnostics, performed mainly in specialistic hospitals, which may make non-specialistic hospital pathology labs eligible for complete formalin-free tissue processing. Even in labs performing liver and kidney diagnostics, reticulin staining can be successfully performed after dipping the slide in just 10 ml of formalin (4% formaldehyde), which can be reused, whereas processing tissues as FFPE requires a tenfold excess volume of 4% formaldehyde for 24 hr incubation for each tissue and approximately 1000 litres 4% formaldehyde a year in a conventional processing machine for our hospital with a catchment area of approximately 700.000 people. This is a reduction in formalin (4% formaldehyde) of over 99% for this process.

Of the thirty-one immunohistochemical markers, twenty-eight gave good or acceptable results with the standard protocol or after optimization of protocols (Table 5 and Fig 4 and S2 and S3). Three of the immunohistochemistry stainings tested did not yet achieve acceptable staining results. Immunohistochemistry antibodies used for diagnostics were selected based on recommendations by the main quality assurance platform for immunohistochemistry used internationally in Europe, Nordiqc (https://www.nordiqc.org). For the three antibodies that were not satisfactory for the diagnostic evaluation on NFPE material, several alternatives are available. CD68 is used for the diagnosis of fibrous-histiocytic tumors, (Langerhans) histiocytosis, and other diseases with an abundance of macrophages and histiocytes. We used antibody KP-1; for CD68 staining, 514H12 and PG are alternative Nordiqc-recommended antibodies (https://www.nordiqc.org). PAX8 is a marker for cells derived from the urologic tract or ovarian origin, and is widely used for the diagnosis of renal cell carcinoma and ovarian non-mucinous carcinoma. We used antibody SP348, but PAX8 antibodies; ZR-1, MXR013, GR002, QR016, or RM436 are alternative Nordiqc recommended antibodies (https://www.nordiqc.org). TDT is a marker used for lymphoma and thymoma diagnostics. For TDT, we used the Ventana RTU polyclonal antibody. Two monoclonal antibodies, SEN28 and EP266 are alternative Nordiqc TDT recommended antibodies (https://www.nordiqc.org). Alternatively, other antibodies currently only used for research purposes could be tested as alternatives, such as MultiMab®, D4B9C of E3O7V for CD68; D2S2I or BC12 for PAX8 (Cell signaling technology®), and for TDT, the IHC771 antibody (GenomeMe Lab Inc.). If necessary, antibodies can be custom made for NFPE tissues if all currently commercially available antibodies do not achieve good results.

Since the immunohistochemistry antibodies combined with the immunofluorescence antibodies used in this study, a total of thirty-five of thirty-eight antibodies gave acceptable results in NFPE tissue stainings, it is most likely that it is possible to obtain or generate working antibodies for all necessary diagnostic and prognostic markers for NFPE material.

We previously showed, using basic tests, that DNA quality may be better preserved in NFPE-processed tissue samples than in FFPE samples [23]. Combined with the findings presented in the current study, this strongly indicates that formalin-free tissue processing is possible without any loss of quality and for many analyses even an improvement for at least four of the most important facets of pathology: histochemistry, immunohistochemistry, immunofluorescence, and molecular pathology.

Despite the observation that good staining results on NFPE for a large amount of histochemistry, immunohistochemistry and immunofluorescence stainings we achieved, we are the first to suggest that embedding tissues without formalin by using the method based on supercritical $CO_2$ may be a better way to produce embedded tissues than the standard formalin-based method that has been used globally by virtually all pathology labs for over 100 years. Broader validation of the NFPE tissues across multiple pathology laboratories would be important to verify our claims and increase the formalin-free pathology staining success rate to 100%.

## Conclusion

With the intention to eliminate the use of the toxic solvent xylene and carcinogenic formaldehyde from the tissue embedding process, a novel method, using supercritical $CO_2$, was used to produce paraffin embedded tissues. Histochemistry, immunohistochemistry, and immunofluorescence stainings used for diagnostics in pathology were assessed. The vast majority, 45 out of 49 stainings (92%), gave good results without using formaldehyde. The requirement of a formalin dip

for reticulin staining does currently undermine a completely formalin-free process. For the other stainings, we show that tissue processing can be done without toxic xylene and carcinogenic formalin (NFPE) and that it produces high-quality samples, suitable for histochemistry, immunohistochemistry, and immunofluorescence.

## Supporting information

**S1 Fig. Supplementary histochemistry stainings.**
(ZIP)

**S2 Fig. Supplementary immunohistochemistry stainings.**
(ZIP)

**S3 Fig. Supplementary immunohistochemistry stainings.**
(ZIP)

**S4 Fig. Supplementary immunofluorescence stainings.**
(ZIP)

**S5 Fig. Quantification of P63 staining intensity.**
(PDF)

## Author contributions

**Conceptualization:** Agnes Marije Hoogland.

**Data curation:** Maarten Niemantsverdriet, Agnes Marije Hoogland.

**Formal analysis:** Maarten Niemantsverdriet, Agnes Marije Hoogland.

**Investigation:** Anne Mei Eshuis, Naomi van der Horst, Rory Ogink, Pien van Smeerdijk.

**Methodology:** José van der Starre-Gaal, Agnes Marije Hoogland.

**Supervision:** José van der Starre-Gaal, Agnes Marije Hoogland.

**Writing – original draft:** Maarten Niemantsverdriet.

**Writing – review & editing:** Agnes Marije Hoogland.

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
