## [Decision Letter · Decision Letter 0]

16 Oct 2025

PONE-D-25-46295High-quality histochemistry, immunohistochemistry, and immunofluorescence on xylene- and formalin-free paraffin-embedded tissuesPLOS ONE

Dear Dr. Hoogland,

Thank you for submitting your manuscript to PLOS ONE. After careful consideration, we feel that it has merit but does not fully meet PLOS ONE’s publication criteria as it currently stands. Therefore, we invite you to submit a revised version of the manuscript that addresses the points raised during the review process.

In your revised manuscript, please ensure that you address the following concerns raised by Reviewers 1 and 2. While both reviewers agree that your study presents an innovative, well-written, and highly relevant advance in formalin- and xylene-free tissue processing, several revisions are necessary to strengthen methodological rigor, reproducibility, and quantitative support for your conclusions. While you should address all reviewer comments completely, I will point out the following critical comments:

Specify the number of biological and technical replicates analyzed for each staining approach.Identify who performed morphological assessments (e.g., pathologists, trained reviewers) and state whether evaluations were blinded.Include counts or percentages of marker-positive cells (e.g., P63⁺), signal intensity measures, or reproducibility scoring.Include appropriate statistical analyses to support statements of equivalence or similarity between FFPE, NFPE, and XFPE groups.Discuss the diagnostic implications of markers that failed in NFPE (CD68, PAX-8, TDT) and note possible solutions such as alternative antibody clones.Provide commentary on the altered collagen morphology observed in NFPE skin samples, and if possible, evaluate whether picrosirius red with polarized light can distinguish collagen subtypes in NFPE-processed tissues.Improve your methods section to provide sufficient methodological detail to allow replication in other laboratories.

We look forward to receiving your revised manuscript.

Kind regards,

Jordan Robin Yaron, Ph.D.

Academic Editor

PLOS ONE

Journal Requirements:

I have read the journal's policy and the authors of this manuscript have the following competing interests:

Agnes Marije Hoogland is included as one of the inventors in the Tispa patent WO 2024/117901 A1 of Tispa Medical B.V. and Stichting Isala Klinieken on the Non-Fixed Paraffin-Embedded (NFPE) technique, with additional patents pending. The other authors do not report a conflict of interest.

3. Please amend the manuscript submission data (via Edit Submission) to include author Marije Hoogland

4. Please amend your authorship list in your manuscript file to include author Agnes Marije Hoogland

Reviewers' comments:

Reviewer's Responses to Questions

**Comments to the Author**

1. Is the manuscript technically sound, and do the data support the conclusions?

Reviewer #1: Yes

Reviewer #2: Partly

2. Has the statistical analysis been performed appropriately and rigorously? 

Reviewer #1: Yes

Reviewer #2: No

3. Have the authors made all data underlying the findings in their manuscript fully available?

Reviewer #1: Yes

Reviewer #2: Yes

4. Is the manuscript presented in an intelligible fashion and written in standard English?

Reviewer #1: Yes

Reviewer #2: Yes

5. Review Comments to the Author

Reviewer #1: This manuscript presents a novel Non-Fixed Paraffin Embedding (NFPE) method for pathological tissues that eliminates the use of both formalin and xylene, instead utilizing supercritical CO₂. The authors conduct a comprehensive evaluation of this technique across histochemistry, immunohistochemistry, and immunofluorescence, using a wide array of markers. The study is timely, relevant, and has significant implications for laboratory safety, environmental sustainability, and molecular diagnostics.

**Strengths:**

- The work clearly represents original research. While prior studies have explored either xylene-free or formalin-free approaches, this paper uniquely integrates both methods and validates the approach across a broad range of stains.

- The breadth of testing is impressive, encompassing 11 histochemical stains, 31 immunohistochemical markers, and 7 immunofluorescence markers.

- The results are generally strong: morphology is well-preserved, DNA quality is likely enhanced, and most markers functioned well with minimal protocol adjustments.

- The clinical and public health relevance is high, as the study provides a pathway to reduce exposure for pathology staff to toxic and carcinogenic substances.

**Concerns / Limitations:**

- **Marker Gaps:** Three immunohistochemical markers (CD68, PAX-8, TDT) did not produce reliable staining with NFPE. This limitation should be explicitly discussed in terms of diagnostic consequences and potential solutions, such as alternative antibody clones.

- **Residual Formalin Use:** The reticulin staining required a brief dip in formalin. While this amount is minimal compared to traditional FFPE workflows, it undermines the claim of a completely formalin-free process. This nuance should be highlighted in the conclusions.

- **Statistical Analysis:** The evaluation is primarily descriptive and based on visual assessment. Incorporating quantitative scoring (e.g., inter-observer agreement, reproducibility measures, signal-to-noise quantification) would strengthen the robustness of the findings.

- **Protocol Reproducibility:** While the authors note that protocols must be optimized for each laboratory, providing more detailed methodological information would assist other labs in reproducing and validating NFPE more effectively.

- **Generalizability:** The study was conducted at a single institution. Broader validation across multiple pathology laboratories would be important before clinical adoption.

**Ethics, Integrity, and Reporting:**

- Ethical standards were met; residual anonymized tissue was utilized with confirmation from the ethics committee.

- The manuscript is well-written in clear English and is easy to follow.

- Data availability complies with journal standards, with supplementary data provided.

**Overall Recommendation:**

This is an important and highly original contribution that advances the field of pathology towards safer and more sustainable practices. With minor revisions—particularly clarifying limitations, expanding the discussion on marker gaps, and enhancing details on reproducibility—the manuscript will be a strong candidate for publication.

Reviewer #2: Niemantsverdriet et al. developed a novel method of tissue processing that eliminates the need for toxic chemicals xylene and formalin. They compared the efficacy of staining in non-formalin fixed tissues to the tissues processed with standard formalin-fixed methods. They tested variety of stains in different organs and they claim that they found comparable results in most of them. Overall, the manuscript is structured well. The methods need more detail, quantifiable parameters need to be used, and statistical analyses need to be performed to compare different groups. Because of lack of quantifiable measures, the conclusions drawn cannot be supported with data, despite histological images showing similar morphology. I have following questions and recommendations:

1. It is recommended that authors describe the NFPE processing method in this manuscript as well.

2. Visual appearance of histological images stained with two different methods need not draw the same conclusions. I suggest adding a quantifiable metric to compare different methods. For example, do you see similar number of P63 positive cells in a tissue processed with FFPE and NFPE? Alternatively, you can develop a visual analog scale and have a blind reviewer assess the histological images of tissues with pathological conditions and confirm if similar inferences are made.

3. The authors should mention who drew the conclusions ‘Good morphology’. Were the conclusions drawn in a double-blinded manner by independent reviewers? Were they experts in identifying the morphology of the tissues? Were they pathologists?

4. In NFPE H&E Skin images, collagen fibrils seem to have lost morphology. Is it consistent in all replicates? Will this technique (NFPE) work with picrosirius red staining with polarized light imaging? Will it be able to separate thin (green) and thicker (red) bundles?

5. I could not see information about the replicates. How many replicates were included per group?

6. There needs to be more discussion about XFPE. Clearly mention the significance and relevance of this group.

6. PLOS authors have the option to publish the peer review history of their article (what does this mean? ). If published, this will include your full peer review and any attached files.

**Do you want your identity to be public for this peer review?** For information about this choice, including consent withdrawal, please see our Privacy Policy .

Reviewer #1: **Yes:** SADIK BAY

Reviewer #2: No

---

## [Author Response · Author response to Decision Letter 1]

4 Dec 2025

Dear Editor,

The previous mail, concerning two issues about informed consent and about our supplementary data is copied below, within this text our reply on these two issues. The bottom text is the earlier reply to the reviewers and comments.

Kind regards,

Agnes Marije Hoogland

PONE-D-25-46295R1

High-quality histochemistry, immunohistochemistry, and immunofluorescence on xylene- and formalin-free paraffin-embedded tissues

Ms Agnes Marije Hoogland

Dear Dr. Hoogland,

We've checked your submission and before we can proceed, we need you to address the following issues:

1. We are unable to open your Supporting Information file " Supplementary data_S1.7z ; Supplementary data_S2-1.7z ; Supplementary data_S2-2.7z ; Supplementary data_S3.7z ". Please kindly revise as necessary and re-upload.

- We have checked and re-uploaded the requested supplementary data.

2. Please provide additional details regarding participant consent. In the Methods section, please ensure that you have specified (1) whether consent was informed and (2) what type you obtained (for instance, written or verbal). If your study included minors, state whether you obtained consent from parents or guardians. If the need for consent was waived by the ethics committee, please include this information.

-We have added the statement on informed consent in the Method section concerning Ethics. The added text is: "The need for informed consent documentation was waived and all experimental protocols were approved as part of the ethics committee approval."

We've returned your manuscript to your account. Please resolve these issues and resubmit your manuscript within 21 days. If you need more time, please email the journal office at plosone@plos.org. We are happy to grant extensions of up to one month past this due date. If we do not hear from you within 21 days, we will withdraw your manuscript.

Please log on to PLOS Editorial Manager at https://www.editorialmanager.com/pone/ to access your manuscript. You will find your manuscript in the 'Submissions Sent Back to Author' link under the New Submissions menu. Be sure to remove your previous manuscript file if you are uploading a new file in response to these requests. After you've made the changes requested above, please be sure to view and approve the revised PDF after rebuilding the PDF to complete the resubmission process.

We are requesting these changes to comply with the PLOS ONE submission guidelines (https://journals.plos.org/plosone/s/submission-guidelines). Please note that we won't send your manuscript for review until you have resolved the above requests.

Thank you for submitting your work to PLOS ONE and supporting our mission of Open Science.

PONE-D-25-46295

High-quality histochemistry, immunohistochemistry, and immunofluorescence on xylene- and formalin-free paraffin-embedded tissues

PLOS ONE

Dear Dr. Hoogland,

Thank you for submitting your manuscript to PLOS ONE. After careful consideration, we feel that it has merit but does not fully meet PLOS ONE’s publication criteria as it currently stands. Therefore, we invite you to submit a revised version of the manuscript that addresses the points raised during the review process.

In your revised manuscript, please ensure that you address the following concerns raised by Reviewers 1 and 2. While both reviewers agree that your study presents an innovative, well-written, and highly relevant advance in formalin- and xylene-free tissue processing, several revisions are necessary to strengthen methodological rigor, reproducibility, and quantitative support for your conclusions. While you should address all reviewer comments completely, I will point out the following critical comments:

Specify the number of biological and technical replicates analyzed for each staining approach.

A: The number of biological replicates is the number of different tissues used as specified in tables 4, 5 and 6. For each tissue, we performed 2 technical replicates after optimization. We now also explain this in the methods section of the revised manuscript, and mention this in the caption of the tables.

Identify who performed morphological assessments (e.g., pathologists, trained reviewers) and state whether evaluations were blinded.

A: The assessments of all stainings were initially performed blinded by an expert reviewer (specialized technician), followed by a not-blinded assessment by a general pathologist and finally verified by a specialized pathologist. We now explain this in the methods section of the revised manuscript and mention this in the caption of tables 4, 5, and 6.

Include counts or percentages of marker-positive cells (e.g., P63⁺), signal intensity measures, or reproducibility scoring.

A: We quantified P63 signal intensity scoring in FFPE, XFPE, and NFPE stainings as requested and show in a new supplementary figure (S4) that we were able to obtain P63 staining intensities in NFPE, XFPE and FFPE tissues that show no statistical difference.

Include appropriate statistical analyses to support statements of equivalence or similarity between FFPE, NFPE, and XFPE groups.

A: As mentioned above, we quantified P63 signal intensity scoring between FFPE, NFPE, and XFPE groups and showed that we were able to achieve signal intensities that show no statistical difference. Observations in this manuscript were initially performed double-blinded by an expert reviewer (specialized technician) and verified (not-blinded) by both a general pathologist and a specialized pathologist. We achieved a 100% inter-observer agreement and we now describe that in the revised manuscript and mention this in the caption of tables 4, 5, and 6. We now more clearly describe in the revised manuscript why we used observational analysis as preferred method for assessment of stainings. We added a new reference (Elliot et. all.) describing that because tissues lack homogeneity, already in unprocessed form, the resulting stainings also lack homogeneity, this is independent of tissue processing method. Therefore, for a general, novel study, testing a large quantity of histochemical, immunohistochemical, and immunofluorescence stainings, thorough observational analysis is more appropriate than statistical comparison.

Discuss the diagnostic implications of markers that failed in NFPE (CD68, PAX-8, TDT) and note possible solutions such as alternative antibody clones.

A: In the revised manuscript, we discuss diagnostic implications of the markers that failed and note possible solutions.

Provide commentary on the altered collagen morphology observed in NFPE skin samples, and if possible, evaluate whether picrosirius red with polarized light can distinguish collagen subtypes in NFPE-processed tissues.

A: In our laboratory and in the other diagnostic pathology laboratories in the Netherlands, the picrosirius red staining is not used for diagnostics. To assess collagen morphology in diagnostics, we use Azan, Masson Goldner (trichrome) and Elastica van Gieson (EvG) stainings. We performed these stainings on other tissue types in our original manuscript, but not yet on skin NFPE. In the revised manuscript, we performed these stainings on skin NFPE samples and added a new figure (Figure 2 in the revised manuscript). We describe the findings in a new paragraph. We did not observe altered (collagen) morphology.

Improve your methods section to provide sufficient methodological detail to allow replication in other laboratories.

A: We have now extended the methods section in the revised manuscript and explained the NFPE process in detail.

We look forward to receiving your revised manuscript.

Kind regards,

Jordan Robin Yaron, Ph.D.

Academic Editor

PLOS ONE

Journal Requirements:

I have read the journal's policy and the authors of this manuscript have the following competing interests:

Agnes Marije Hoogland is included as one of the inventors in the Tispa patent WO 2024/117901 A1 of Tispa Medical B.V. and Stichting Isala Klinieken on the Non-Fixed Paraffin-Embedded (NFPE) technique, with additional patents pending. The other authors do not report a conflict of interest.

A: We have now included the statement in the cover letter and in the revised manuscript.

3. Please amend the manuscript submission data (via Edit Submission) to include author Marije Hoogland

A: We added Agnes Marije Hoogland to the manuscript submission data

4. Please amend your authorship list in your manuscript file to include author Agnes Marije Hoogland

A: We have now added Agnes to Marije Hoogland’s name in the revised manuscript.

A: We have now added captions for supporting information files in the revised manuscript.

A: There was no specific recommendation to cite previously published work

Reviewers' comments:

Reviewer's Responses to Questions

Comments to the Author

1. Is the manuscript technically sound, and do the data support the conclusions?

Reviewer #1: Yes

Reviewer #2: Partly

2. Has the statistical analysis been performed appropriately and rigorously?

Reviewer #1: Yes

Reviewer #2: No

3. Have the authors made all data underlying the findings in their manuscript fully available?

Reviewer #1: Yes

Reviewer #2: Yes

4. Is the manuscript presented in an intelligible fashion and written in standard English?

Reviewer #1: Yes

Reviewer #2: Yes

5. Review Comments to the Author

Reviewer #1: This manuscript presents a novel Non-Fixed Paraffin Embedding (NFPE) method for pathological tissues that eliminates the use of both formalin and xylene, instead utilizing supercritical CO₂. The authors conduct a comprehensive evaluation of this technique across histochemistry, immunohistochemistry, and immunofluorescence, using a wide array of markers. The study is timely, relevant, and has significant implications for laboratory safety, environmental sustainability, and molecular diagnostics.

**Strengths:**

- The work clearly represents original research. While prior studies have explored either xylene-free or formalin-free approaches, this paper uniquely integrates both methods and validates the approach across a broad range of stains.

- The breadth of testing is impressive, encompassing 11 histochemical stains, 31 immunohistochemical markers, and 7 immunofluorescence markers.

- The results are generally strong: morphology is well-preserved, DNA quality is likely enhanced, and most markers functioned well with minimal protocol adjustments.

- The clinical and public health relevance is high, as the study provides a pathway to reduce exposure for pathology staff to toxic and carcinogenic substances.

**Concerns / Limitations:**

- **Marker Gaps:** Three immunohistochemical markers (CD68, PAX-8, TDT) did not produce reliable staining with NFPE. This limitation should be explicitly discussed in terms of diagnostic consequences and potential solutions, such as alternative antibody clones.

A: We now d

---

## [Decision Letter · Decision Letter 1]

23 Dec 2025

High-quality histochemistry, immunohistochemistry, and immunofluorescence on xylene- and formalin-free paraffin-embedded tissues

PONE-D-25-46295R1

Dear Dr. Hoogland,

We’re pleased to inform you that your manuscript has been judged scientifically suitable for publication and will be formally accepted for publication once it meets all outstanding technical requirements.

Kind regards,

Jordan Robin Yaron, Ph.D.

Academic Editor

PLOS One

Additional Editor Comments (optional):

Reviewers' comments:

Reviewer's Responses to Questions

**Comments to the Author**

1. If the authors have adequately addressed your comments raised in a previous round of review and you feel that this manuscript is now acceptable for publication, you may indicate that here to bypass the “Comments to the Author” section, enter your conflict of interest statement in the “Confidential to Editor” section, and submit your "Accept" recommendation.

Reviewer #1: All comments have been addressed

Reviewer #2: All comments have been addressed

2. Is the manuscript technically sound, and do the data support the conclusions?

Reviewer #1: Yes

Reviewer #2: Yes

3. Has the statistical analysis been performed appropriately and rigorously? 

Reviewer #1: Yes

Reviewer #2: Yes

4. Have the authors made all data underlying the findings in their manuscript fully available?

Reviewer #1: Yes

Reviewer #2: Yes

5. Is the manuscript presented in an intelligible fashion and written in standard English?

Reviewer #1: Yes

Reviewer #2: Yes

6. Review Comments to the Author

Reviewer #1: The authors have been improved the manuscript as my suggestions. I am happy to advice to accept the manuscript

Reviewer #2: The authors have addressed all the comments satisfactorily. I think authors need to discuss the quantification results (Figure S4) in the "Results" section.

7. PLOS authors have the option to publish the peer review history of their article (what does this mean? ). If published, this will include your full peer review and any attached files.

**Do you want your identity to be public for this peer review?** For information about this choice, including consent withdrawal, please see our Privacy Policy .

Reviewer #1: **Yes:** SADIK BAY

Reviewer #2: No

---

## [Editor Report · Acceptance letter]

PONE-D-25-46295R1

PLOS One

Dear Dr. Hoogland,

I'm pleased to inform you that your manuscript has been deemed suitable for publication in PLOS One. Congratulations! Your manuscript is now being handed over to our production team.

Kind regards,

on behalf of

Dr. Jordan Robin Yaron

Academic Editor

PLOS One